# OpenReview forum: "Explainable Reinforcement Learning from Human Feedback to Improve Alignment"
_NeurIPS.cc/2025/Conference — NeurIPS 2025 poster_

### Official Review · Reviewer_66Se · 2025-06-30

**Clarity:** 2
**Significance:** 3
**Originality:** 3
**Rating:** 4
**Confidence:** 4

**Summary:**

This paper introduces a method to further improve the effectiveness of RLHF methods after they are finished. It focuses on improving the performance on the part of the validation dataset where the model did not perform well. This is done by first identifying the training data (in the binary preference dataset) which are most similar to each problematic prompt-response pair, then performing some unlearning in the reward model and some fine-tuning for the policy model. Experiments show overall better win rates of the proposed approach over existing RLHF approaches (PPO, REMAX) separately on two datasets. Performance tradeoffs are observed on the non-problematic part of the validation dataset.

**Questions:**

1. It is claimed that the method finds the causes of why the policy model generated unsatisfactory responses. What is your definition of a cause here? There are the research fields on explainable AI and causality / actual causes. How does your definition fit in?
2. Why does identifying relevant examples in the preference dataset, which was only used to train the reward models and not directly the policy model, counts as explaining the unsatisfactory generations?
3. Regarding the approach for separating the evaluation dataset into satisfactory and unsatisfactory subsets – how valid is setting a threshold using a reward model for this purpose? In line 608 you mentioned “the score lower bound of the satisfactory group is usually above the score upper bound of the unsatisfactory group” which does not seem clear. My intuitions are that it is difficult to compare the scalar outputs of the reward models for responses across different input prompts, because the prompts themselves will influence the magnitude of the scores?
4. How's the reward models' performance after unlearning?

**Ethical Concerns:**

["NO or VERY MINOR ethics concerns only"]

**Final Justification:**

I increased my score from 2 to 4 following the rebuttal which addressed my concerns about the clarity of being "explainable" and about the significance of results given the reward model performance after unlearning. That said, the inclusion of these additional discussions and experiments to the main text might not be trivial.

**Limitations:**

All parts about explainability should be revisited with clear discussions on motivations and assumptions. The evaluations are not comprehensive enough to support the relevant contributions made in the paper. See detailed comments on weaknesses and questions. I think those are essential aspects to elaborate on in the paper's main text.

**Quality:**

3

**Strengths And Weaknesses:**

Strengths:
1. The paper is well-presented with clear problem definitions and good visualisations. The mathematical formulations are also very easy to follow.
2. The methodology sections are sound and well-explained.

Weaknesses:
1. Clarity and Significance: The paper claims that the proposed approach is explainable. However, discussions on explainability are very shallow. The most relevant part might be lines 138-144, but these motivations hint that the method is aiming to explain why the reward model assigns a high reward to the policy’s response, rather than to explain why the policy LLM generates such a response. There are also no real evaluations for explainability, as all evaluations are regarding win rates. Therefore, using the word explainable to describe this paper is not appropriate.

2. Clarity and Significance: Motivations and assumptions are lacking for the method. As mentioned in the paper, LLM post-training involves SFT, learning a reward model, and running RL. However, the proposed approach deems that the cause of unsatisfactory LLM behaviour lies in the preference training data for the reward model. This dataset is usually not directly seen by the LLM itself, so it is questionable to ignore the other data sources where the reasons why the LLM unsatisfactory behaviour might exist. This is a fundamental aspect to consider, but it is not discussed in the paper. (While in the experiments the three steps use the same dataset, lacking such discussions will also harm the generalisability, hence the significance of the proposed approach.)

3. Quality: Writing (apart from the mathematical formulations, those are good) is often too simplistic or unconvincing. For example, in abstract and intro, discussions on *cause* are very vague; in Line 25, one should not expect a model to become all perfect after some training; in Preliminary section, high-level descriptions for the equations behind RM and RL would be helpful. Conclusions and related works are very incomplete; future works and limitations are not well-discussed.

4. Quality and Significance: Evaluations are thin and a bit too narrow.
(1) Only 2 datasets and 3 small models are involved.
(2) The choices of the open-source reward models used to evaluate LLM responses are not well-justified – neither is state of the art, especially the deberta model. It’d be good to refer to the rewardbench leaderboard to find reasonably sized and good ones. While GPT-4 is also used, it is suggested that the order in which the candidate responses are presented be randomised – currently in the code, it’s always showing the response by the proposed method first and then the baseline response (if I’m not mistaken).
(3) The scope of evaluation is narrow, as only the win rate metric is used on only one dataset in each experiment. It is not clear whether the proposed RLHF approach worsens the LLMs’ general performance, e.g., on other benchmarks, especially as some unlearning is involved.

---

> ### Author Rebuttal · Authors · 2025-07-24
>
> Thanks for your constructive reviews. We address your comments in a point-to-point manner.
>
> **Answer to Weakness 1**: Thanks for mentioning the explanation. Please note that the primary focus of this paper is to improve LM performance, and the explanation module is used as a tool to achieve this goal. Therefore, we evaluate the effectiveness of our method via downstream win rates.
> Moreover, we provide a mathematically grounded explanation module and evaluate the explanation in subsection 6.3. First, our explanation follows the definition of post-hoc example-based explanation in [D1,D2] where a model’s decision is explained by identifying relevant examples from training data. In our setting, we find a set of training data to explain why a test-time response $\bar{y}$ is generated by explaining why $(\bar{x},\bar{y})$ has a high reward. This reformulation is mathematically grounded: the identified examples contribute to the high value of $r_\theta(\bar{x},\bar{y})$, and since RL optimizes the LM to maximize reward, a high reward $r_\theta(\bar{x},\bar{y})$ increases the likelihood that $\bar{y}$ is generated. Therefore, the identified examples (indirectly) contribute to the generation of $\bar{y}$. Second, we explicitly evaluate the fidelity (a widely used metric in XRL) of the explanation in subsection 6.3. In the case of explaining why the model generates unsatisfactory responses, fidelity in XRL [D3,D4] is defined as performance improvement from acting on the explanation. In our case, it means whether the unlearned model generates better responses. We measure this improvement using win rates of the unlearned model over the original RLHF model. Since the win rate is larger than 50\%, it means that the unlearned model generates better responses. We also provide qualitative explanation examples in Appendix D.2, which is human-interpretable.
> Following your suggestion, we add a qualitative evaluation based on human plausibility, which is a widely-used metric in XRL [D3,D4] that directly evaluates the explanation. We recruited 15 participants, and for each test-time response, provided them with our explanation and a set of randomly selected training examples of the same size. Participants were asked to choose which set better helps them understand why the model generates the response. The percentages favoring our explanation in the four tasks (opt+full-hh-rlhf, pythia+full-hh-rlhf, pythia+tldr, llama+tldr) are: 87%, 93%, 80%, 80%.
>
> **Answer to Weakness 2**: We would like to clarify that the goal of the explanation module is to find "a cause" (lines 2 and 40) instead of the sole cause of unsatisfactory responses. We agree that other sources, including SFT and RL, can also contribute to unsatisfactory responses. However, it is infeasible to enumerate all such causes, and this paper focuses on the reward data for explanation. In our experiments, the three stages use disjoint partitions of the dataset (line 278). Thus, the LM does not directly observe the data used to train the reward model. However, focusing on the reward data remains well-justified: although the LM does not directly access this data, this data is used to train the reward model, and RL optimizes the LM to maximize the reward. Therefore, the reward data indirectly but causally influences the LM’s responses. We will include this discussion to clarify this causal pathway and acknowledge other possible sources of causes.
>
> **Answer to Weakness 3**: Following your suggestion, we will clarify the causal reasoning path in our setting: reward training data influences the reward model, which determines reward values for LM outputs, which in turn shapes the LM policy via RL optimization. Therefore, examples that contribute to a high reward for a response indirectly but causally increase the likelihood that the LM generates that response. We will rephrase line 25 to emphasize "many responses are unsatisfactory after RLHF". We will add brief description on RM (which learns a reward signal to approximate human preference) and RL (which learns an LM to optimize reward). We have additional complete related works in Appendix A. We will include the limitation that this paper does not address other sources of causes, and one future work is to find other sources of causes.
>
> **Answer to Weakness 4**: Following your suggestion, we add the following results:
>
> 1. We conduct an additional experiment using llama-2-13b model and imdb dataset. We report the win rates over the SFT model. The win rates are evaluated by Skywork-Reward-V2-Llama-3.1-8B (ranked top in RewardBench 2) and GPT-4 with the order of the candidate responses randomized:
>
> |                                | PPO  | PPO+XRLHF | ReMax | ReMax+XRLHF |
> |--------------|------|-----------|--------|--------------|
> | Skywork-Reward-V2-Llama-3.1-8B | 66.5 | 71.4      | 67.1   | 70.8         |
> | GPT-4                          | 69.6 | 72.5      | 70.7   | 73.2         |
>
> The above results demonstrate that our method (XRLHF) can further improve the base RLHF algorithms (PPO and ReMax) on a large base model (llama-2-13b) and a new dataset (imdb).
>
> 2. We evalute the models on new and unseen datasets. Recall that the experiment section has two tasks: dialogue and summarization. The dialogue task includes opt-1.3B and pythia-2.8B models trained on the full-hh-rlhf dataset. The summarization task includes pythia-2.8B and llama-2-7B models trained on the Reddit TL;DR dataset. Here we evaluate the models trained in the dialogue task on a new dataset PKU-SafeRLHF, and evaluate models trained in the summarization task on a new dataset webis/tldr-17.
> (1) opt dialogue task
>
> |                                | PPO  | PPO+XRLHF | ReMax | ReMax+XRLHF |
> |-----|------|-----------|--------|--------------|
> | Skywork-Reward-V2-Llama-3.1-8B | 64.2 | 68.0      | 66.4   | 71.4         |
> | GPT-4                          | 66.3 | 68.5      | 65.1   | 69.7         |
>
> (2) pythia dialogue task
>
> |                                | PPO  | PPO+XRLHF | ReMax | ReMax+XRLHF |
> |-|-|-----------|--------|--------------|
> | Skywork-Reward-V2-Llama-3.1-8B | 62.5 | 64.1      | 60.7   | 64.6         |
> | GPT-4                          | 66.1 | 66.8      | 62.4   | 67.1         |
>
> (3) pythia summarization task
>
> |                                | PPO  | PPO+XRLHF | ReMax | ReMax+XRLHF |
> |--------------------------------|------|-----------|--------|--------------|
> | Skywork-Reward-V2-Llama-3.1-8B | 60.6 | 67.2      | 62.2   | 62.4         |
> | GPT-4                          | 59.8 | 64.4      | 67.9   | 69.3         |
>
> (4) llama summarization task
>
> |                                | PPO  | PPO+XRLHF | ReMax | ReMax+XRLHF |
> |--------------------------------|------|-----------|--------|--------------|
> | Skywork-Reward-V2-Llama-3.1-8B | 65.1 | 67.8      | 67.3   | 65.1         |
> | GPT-4                          | 62.7 | 61.5      | 66.8   | 67.9         |
>
> The above results demonstrate that our method can also improve RLHF performance on new and unseen datasets.
>
> **Answer to Question 1**: In our setting, we define a cause as a set of reward training data that indirectly contributes to the LM’s generation of $\bar{y}$ by contributing to the high value of $r_\theta(\bar{x},\bar{y})$. As discussed in our answer to Weakness 1, this causal path is mathematically grounded. Moreover, our explanation follows the definition of post-hoc example-based explanation [D1,D2] where a model's decision is explained by influential training examples. In our case, we identify the training examples to explain why the reward model assigns a high value to $(\bar{x},\bar{y})$, and this further explains why the LM generates $\bar{y}$ through the causal path.
>
> **Answer to Question 2**: Please refer to our answers to Weaknesses 1 and 2.
>
> **Answer to Question 3**: We agree that different prompts can affect the magnitude of the raw reward output by the reward model. Therefore, we compare calibrated rewards, rather than raw rewards, to the reward threshold. Specifically, the calibrated reward of a response is its raw reward subtracting the raw reward of the SFT model response to the same prompt. This calibration reduces prompt-specific bias in reward values. We then apply a fixed threshold to these calibrated rewards to classify satisfactory and unsatisfactory responses. We will clarify this calibration to avoid confusion.
>
> **Answer to Question 4**: We evaluate the reward model’s performance before and after unlearning by measuring its agreement with GPT-4 preferences. Specifically, we randomly sample 500 examples from the test set, where each example consists of a prompt and two responses. For each pair, we compare the reward model's ranking with GPT-4’s judgment and compute the percentage of cases where the reward model assigns a higher score to the response preferred by GPT-4.
>
> |                     | opt+full-hh-rlhf | pythia+full-hh-rlhf | pythia+tldr | llama7b+tldr | llama13b+imdb |
> |---------------------|------------------|----------------------|-------------|---------------|----------------|
> | Before unlearning   | 69.8%            | 72.5%                | 64.4%       | 68.5%         | 77.6%          |
> | After unlearning    | 71.6%            | 74.3%                | 64.1%       | 70.4%         | 80.2%          |
>
> The above results  show that unlearned reward has better performance than the original reward.
>
> **Answer to Limitation**: Following your suggestion, we have addressed all your comments in our answers to the weaknesses and questions.
>
> [D1] J.Crabbé et al, "Explaining latent representations with a corpus of examples", NeurIPS, 2021.
>
> [D2] H Sun et al, "Accountability in offline reinforcement learning: Explaining decisions with a corpus of examples", NeurIPS, 2024
>
> [D3] W Guo et al, "Edge: Explaining deep reinforcement learning policies", NeurIPS, 2021.
>
> [D4] Z Cheng et al, "Rice: Breaking through the training bottlenecks of reinforcement learning with explanation", ICML, 2024.

---

> > ### Comment · Reviewer_66Se · 2025-08-05
> > **Response to author rebuttal**
> >
> > Thank you for the very detailed response to my concerns. I still think the title and abstract are a bit misleading, as they really raise expectations regarding the level of explainability achieved in this work. Explicitly explaining the causal path (as mentioned in the rebuttal) would be essential to improve the paper's clarity. The reward model performance experiments would also be important to complement the current main results in the paper.
> >
> > Overall, my concerns have been addressed, and I am willing to raise my score.

---

> ### Author Response · Authors · 2025-08-05
>
> We sincerely appreciate your time and effort in reviewing our paper. Thank you for recognizing the contributions of our work and for raising the score.
>
> Following your suggestion, we will revise the title and abstract to emphasize that this paper uses explanation as a tool to improvement alignment, rather than focusing on the explanation itself. We will highlight the causal path to calrify why focusing on the reward training data is reasonable to explain the LM's response. Additionally, we will include the evaluations on the reward model performance in the experiment section.
>
> Thank you again for acknowledging our efforts in addressing your concerns.

---

### Official Review · Reviewer_8SuZ · 2025-07-01

**Clarity:** 3
**Significance:** 2
**Originality:** 3
**Rating:** 5
**Confidence:** 4

**Summary:**

This paper presents a method , named Explainable Reinforcement Learning from Human Feedback (XRLHF), which operates as a post-training step for RLHF and is inspired by the human strategy of identifying and correcting the cause of an undesirable outcome.

The paper's primary contributions are the novel formulation of explaining unsatisfactory responses by identifying causative training data and the two-phase unlearning method to improve the model without complete retraining.

**Questions:**

1. Regarding Table 4, on which types of satisfactory samples does the model's performance degrade? Is this trade-off controllable?
2. Does the unlearning process affect the model's generalization performance on other benchmarks (e.g. ApacaEval)? And what about performance of XRLHF on Llama3 / Qwen2？
3. Could there be an experiment that uses a reward model to analyze the score distribution of the rejected and chosen answers for the unsatisfactory samples? In my opinion, this is a prerequisite for XRLHF to be effective.
4. Doesn't it seem a bit strange that Algorithm 1 in the paper named XRLHF doesn't contain the RLHF (unlearning) process?

If these concerns can be addressed, I will be glad to increase my rating.

**Ethical Concerns:**

["NO or VERY MINOR ethics concerns only"]

**Final Justification:**

I appreciate the author's efforts in providing more experiments and more discussion on my concerns. I will revise my scores, conditioned on the authors adding these experimental results and discussion to the final version of the paper.

**Limitations:**

Yes

**Quality:**

3

**Strengths And Weaknesses:**

**Strengths**

1. The paper introduces an innovative two-stage "explain-and-correct" framework to identify and fix unsatisfactory responses from language models trained with RLHF. As a post-training method, it offers a highly practical way to improve models that are already deployed or expensive to retrain.
2. The method is technically sound. It creatively formulates the "explanation" step as a constrained optimization problem and designs a tailored "unlearning" strategy for the RLHF pipeline, which first corrects the reward model and then fine-tunes the policy model.
3. The paper is clearly written and well-organized. Figure 1, in particular, intuitively illustrates the entire workflow, which greatly aids reader comprehension.

**Weaknesses:**

1. The essence of Explainable RLHF (XRLHF) is a data filtering mechanism, but the proportion of data being filtered has not been well-studied through ablation.
2. The process of using a reward model to filter the dataset D inherently introduces bias.
3. Regarding Table 4, on which types of satisfactory samples does the model's performance degrade? Is this trade-off controllable?
4. The evaluation is only conducted on an in-distribution test set. Does the unlearning process affect the model's generalization performance on other benchmarks (e.g. ApacaEval)?
5. There is no analysis subsection dedicated to the data filtered by the explainable method. Could there be an experiment that uses a reward model to analyze the score distribution of the rejected and chosen answers for the unsatisfactory samples?

---

> ### Author Rebuttal · Authors · 2025-07-31
>
> Thanks for your constructive reviews. We address your comments in a point-to-point manner.
>
> **Answer to Weakness 1**: We add the following ablation study on the proportion of data used for unlearning. Specifically, for each unsatisfactory response, our method identifies a convex decomposition over a subset of reward training data, assigning a weight to each data in this subset. In our original setup, we selected the top 30\% training data with the highest weights as the explanation and unlearn these data. Here we vary this proportion and unlearn the top 0\%, 10\%, 30\%, 50\%, 70\%, and 100\% data. We report the GPT-4 win rates over the SFT model on the test set. Recall that our method improves over base RLHF algorithms and we conduct this ablation for two base RLHF algorithms, PPO and ReMax.
>
> (1) opt-1.3B + full-hh-rlhf
>
> | Base RLHF algorithm | 0%   | 10%  | 30%  | 50%  | 70%  | 100% |
> |-|--|--|--|--|--|--|
> | PPO                 | 68.0 | 70.1 | 78.5 | 76.5 | 68.4 | 58.8 |
> | ReMax               | 66.9 | 72.8 | 76.1 | 77.2 | 66.2 | 52.1 |
>
> (2) pythia-2.8B + full-hh-rlhf
>
> | Base RLHF algorithm | 0%   | 10%  | 30%  | 50%  | 70%  | 100% |
> |-|--|--|--|--|--|--|
> | PPO                 | 67.5 | 68.8 | 76.5 | 71.5 | 61.2 | 54.2 |
> | ReMax               | 66.8 | 74.2 | 80.1 | 74.4 | 69.4 | 61.5 |
>
> (3) pythia-2.8B + TL;DR
>
> | Base RLHF algorithm | 0%   | 10%  | 30%  | 50%  | 70%  | 100% |
> |-|-|-|-|-|-|-|
> | PPO                 | 70.4 | 75.0 | 79.2 | 68.1 | 66.6 | 62.8 |
> | ReMax               | 71.2 | 77.6 | 80.4 | 73.5 | 68.4 | 59.5 |
>
> (4) llama-2-7B + TL;DR
>
> | Base RLHF algorithm | 0%   | 10%  | 30%  | 50%  | 70%  | 100% |
> |---------------------|------|------|------|------|------|------|
> | PPO                 | 68.4 | 70.1 | 72.5 | 73.3 | 73.0 | 69.3 |
> | ReMax               | 67.9 | 68.2 | 78.1 | 76.8 | 72.9 | 66.1 |
>
> The above results show that the best proportion to unlearn is around 30\%-50\%. Unlearning too little ($\leq$10\%) is not sufficient enough to enhance alignment, while unlearning too much ($\geq$70\%) can degrade performance.
>
> **Answer to Weakness 2**: We agree that a reward model may introduce bias as it cannot 100\% capture human preference. To mitigate this, we use human evaluation to calibrate the filtering threshold (discussed in Appendix D.1). Specifically, we randomly sample 50 prompt-response pairs from D and use human evaluation to classify satisfactory and unsatisfactory responses. For each response, we compute the calibrated reward, which is the raw reward of this response output by the reward model subtracting the raw reward of the SFT model response to the same prompt. This calibration reduces the prompt-dependent bias. We then determine the filtering reward threshold as the upper bound of the calibrated rewards of the unsatisfactory group. Responses with calibrated rewards under this threshold are labeled as unsatisfactory. This approach ensures that the filtering threshold is grounded in human judgment.
>
> **Answer to Weakness 3**: The samples with performance degradation are the samples that are far from the unsatisfactory prompt-response pairs in the representation space. Specifically, let us denote degrading samples as the samples with performance degradation and denote improving samples as the samples with performance improvement. In the experiment of opt-1.3B model and full-hh-rlhf dataset, the average distance between the improving samples and unsatisfactory prompt-response pairs is around 25, while the average distance between the degrading samples and unsatisfactory prompt-response pairs is around 80.
> This improvement-degradation is trade-off controllable. In particular, our policy fine-tuning objective (5) has a KL term to restrict the divergence of the new policy from the original RLHF policy on the satisfactory responses. The coefficient of the KL divergence is $\bar{\beta}$. A larger $\bar{\beta}$ will result in less degradation on the satisfactory responses but also less improvement on the unsatisfactory responses. We conduct an ablation on this coefficient $\bar{\beta}$ where we use opt-1.3B model, full-hh-rlhf dataset, and ReMax base algorithm. We vary the value of $\bar{\beta}$: 0, 0.05, 0.1, 0.5, 1.0, and report GPT win rates over ReMax on both the unsatisfactory set $D_u$ and satisfactory set $D\setminus D_u$.
>
> |                 | 0    | 0.05 | 0.1  | 0.5  | 1.0  |
> |-----------------|------|------|------|------|------|
> | $D_u$           | 82.8 | 73.2 | 62.6 | 55.0 | 51.2 |
> | $D \\setminus D_u$ | 30.2 | 47.5 | 48.9 | 50.8 | 50.2 |
>
> The above results show that a larger $\bar{\beta}$ results in less degradation on the satisfactory responses but also less improvement on the unsatisfactory responses.
>
> **Answer to Weakness 4**: We evaluate our models on ApacaEval, specifically using the "tatsu-lab/alpaca\_eval" dataset. For each prompt (i.e., "instruction") in this dataset, we use our models to generate responses. Recall that we have four experiments: opt-1.3B+full-hh-rlhf, pythia-2.8B+full-hh-rlhf, pythia-2.8B+TL;DR, llama-2-7B+TL;DR, and we consider two base RLHF algorithms: PPO and ReMax. We report the GPT-4 win rates of RLHF (PPO and ReMax) and our method (XRLHF) over the SFT model:
>
> |                             | PPO  | PPO+XRLHF | ReMax | ReMax+XRLHF |
> |-----------------------------|------|-----------|--------|-------------|
> | opt-1.3B+full-hh-rlhf       | 58.8 | 61.5      | 59.1   | 63.7        |
> | pythia-2.8B+full-hh-rlhf    | 62.1 | 66.2      | 60.6   | 68.7        |
> | pythia-2.8B+TL;DR           | 61.8 | 63.9      | 62.4   | 66.6        |
> | llama-2-7B+TL;DR            | 60.2 | 64.5      | 62.1   | 66.2        |
>
> The above results demonstrate that our method (XRLHF) can also improve RLHF on AlpacaEval.
>
> **Answer to Weakness 5**: In our case, we identify a set of training data to explain the unsatisfactory samples. Here, we provide analysis on the chosen and rejected responses of the identified training data. In particular, we use Beaver-7b-v3.0-reward for the full-hh-rlhf dataset and use Deberta-v3-large-v2 for the TL;DR dataset. We provide the average reward of chosen and rejected responses of identified training data in our four experiments (opt+full-hh-rlhf, pythia+full-hh-rlhf, pythia+TL;DR, llama+TL;DR) where the base RL algorithm is ReMax.
>
> |           | opt+full-hh-rlhf | pythia+full-hh-rlhf | pythia+TL;DR | llama+TL;DR |
> |-----------|------------------|----------------------|---------------|--------------|
> | chosen    | 2.05             | 1.85                 | 3.44          | 3.12         |
> | rejected  | 1.89             | 1.79                 | 3.58          | 3.64         |
>
> The above results show that the average reward of rejected responses is very close to chosen responses for identified training data in the full-hh-rlhf dataset, and is even larger than the chosen responses for identified data in the TL;DR dataset. To further inspect the reward distribution of the chosen and rejected responses of the identified training data, we provide the percentage of the identified training data where the rejected response has a higher reward than the chosen response:
>
> | opt+full-hh-rlhf | pythia+full-hh-rlhf | pythia+TL;DR | llama+TL;DR |
> |------------------|----------------------|---------------|--------------|
> | 37%              | 44%                  | 62%           | 58%          |
>
> The above results indicate that the identified training data include a large amount of misleading signal where the rejected response is actually better than the chosen response.
>
> **Answer to Question 1**: Please refer to our answer to Weakness 3.
>
> **Answer to Question 2**: Please refer to our answer to Weakness 4 where we evaluate our models on AlpacaEval. We conduct an additional experiment using Llama-3.1-8B model and full-hh-rlhf dataset for training, and we provide GPT-4 win rates over the SFT model on the alpaca\_eval dataset.
>
> | PPO  | PPO+XRLHF | ReMax | ReMax+XRLHF |
> |------|-----------|--------|-------------|
> | 62.1 | 65.2      | 63.9   | 67.7        |
>
> The above results demonstrate that XRLHF can improve RLHF using Llama-3.1-8B base model.
>
> **Answer to Question 3**: Please refer to our answer to Weakness 5.
>
> **Answer to Question 4**: Our method includes two parts: explanation and improvement. Algorithm 1 (XRLHF) only includes the explanation and thus do not include the improvement part (i.e., unlearning and finetuning). We will add an overall algorithm that includes both parts.

---

> ### Comment · Reviewer_8SuZ · 2025-08-06
> **reply by the reviewer**
>
> I will carefully read the reviewers’ responses and reply to them today. I’ve noticed that the responses are quite extensive.
>
> ---
>
> Update: I appreciate the author's efforts in providing more experiments and more discussion on my concerns. I will revise my scores, conditioned on the authors adding these experimental results and discussion to the final version of the paper.

---

> > ### Author Response · Authors · 2025-08-06
> >
> > We sincerely appreciate your time and effort in reviewing our paper and thank you for recognizing the contributions of this work. Following your suggestion, we will include these additional experiment results in the final version of the paper. We will also add the discussions to clarify how we filter the dataset $D$ and to clarify the trade-off between unsatisfactory response improvement and satisfactory response degradation.
> >
> > Thank you again for recognizing our efforts in addressing your comments.

---

### Official Review · Reviewer_Xdyj · 2025-07-02

**Clarity:** 3
**Significance:** 3
**Originality:** 3
**Rating:** 4
**Confidence:** 4

**Summary:**

This paper presents a method to identify the causes of low-quality responses in RLHF settings. The core idea is to locate the most relevant preference pairs in the reward model training data that are associated with unsatisfactory outputs, and then perform unlearning on these data. Experimental results demonstrate the method’s effectiveness on dialogue generation and summarization tasks.

**Questions:**

1. It is necessary to conduct experiments on more recent models (at least LLaMA-3.1 or Qwen-2.5).

2. This method should also get a better reward model, so its effectiveness need to be validated on reward model benchmarks such as RewardBench, PPE-Preference, RMB, RM-Bench, and JudgeBench.

**Ethical Concerns:**

["NO or VERY MINOR ethics concerns only"]

**Final Justification:**

After reviewing all the current materials, I maintain my original assessment. The lack of comprehensive experiments on recent models of sufficient size is the reason preventing a higher score.

**Limitations:**

Yes.

**Paper Formatting Concerns:**

NA.

**Quality:**

2

**Strengths And Weaknesses:**

Strengths:

1. The paper is clearly written and easy to follow.

2. The problem addressed is highly interesting and important: identifying the root causes of unsatisfactory model outputs and mitigating them through unlearning. The proposed solution is well-motivated and carefully designed.

Weaknesses:

The base model is too weak, and models from two years ago are not sufficiently convincing given the current progress in large language models. Since policy unlearning based on negative gradient updates has already been widely used, the minimum-distance-based iterative training data selection is the main contribution of this paper. However, its effectiveness needs to be demonstrated on more recent and advanced models through broader and more up-to-date experiments.

---

> ### Author Rebuttal · Authors · 2025-07-31
>
> Thanks for your constructive review. We address your comments in a point-to-point manner.
>
> **Weakness**: The base model is too weak, and models from two years ago are not sufficiently convincing given the current progress in large language models. Since policy unlearning based on negative gradient updates has already been widely used, the minimum-distance-based iterative training data selection is the main contribution of this paper. However, its effectiveness needs to be demonstrated on more recent and advanced models through broader and more up-to-date experiments.
> **Answer**: Following your suggestion, we conduct an additional experiment where we use Llama-3.1-8B as the base model and full-hh-rlhf dataset as the training dataset. Same as the main paper, we partition the training data into three parts: 20\% for supervised finetuning, 40\% for reward learning, and 40\% for reinforcement learning. We consider two base RLHF algorithms: PPO and ReMax. We report the GPT-4 win rates of the base RLHF algorithms and our method (XRLHF) over the SFT model on the test set of full-hh-rlhf:
>
> | PPO  | PPO+XRLHF | ReMax | ReMax+XRLHF |
> |-|--|--|--|
> | 69.4 | 75.1      | 71.6  | 77.9        |
>
> The above results demonstrate that our method can further improve RLHF (e.g., PPO and ReMax) using the recent model Llama-3.1-8B.
>
> **Question 1**: It is necessary to conduct experiments on more recent models (at least LLaMA-3.1 or Qwen-2.5).
> **Answer**: Please refer to our answer to Weakness.
>
> **Question 2**: This method should also get a better reward model, so its effectiveness need to be validated on reward model benchmarks such as RewardBench, PPE-Preference, RMB, RM-Bench, and JudgeBench.
> **Answer**: We evaluate our reward models on RewardBench, specifically using the "allenai/reward-bench" dataset. This dataset has two splits: "raw" and "filtered", and we use the "filtered" split. Each data point in this split includes a prompt, a chosen response, and a rejected response. For evaluation, we score both the chosen and rejected responses using our reward models. If the reward assigned to the chosen response is higher than that of the rejected response, the reward model is considered correct at this data example. The overall score is calculated as the percentage of examples where the reward model is correct.
> We provide scores of our reward models (before and after unlearning) across the five experiments: opt-1.3B+full-hh-rlhf, pythia-2.8B+full-hh-rlhf, pythia-2.8B+TL;DR, llama-2-7B+TL;DR, llama-3.1-8B+full-hh-rlhf. Note that we use two different base RLHF algorithms: PPO and Remax. Although these two RLHF algorithms use the same reward model before unlearning, they can generate different unsatisfactory responses and thus lead to different unlearned reward models. Therefore, we report two versions of the unlearned reward models corresponding to PPO and ReMax.
>
> |                          | opt-1.3B+full-hh-rlhf | pythia-2.8B+full-hh-rlhf | pythia-2.8B+TL;DR | llama-2-7B+TL;DR | llama-3.1-8b+full-hh-rlhf |
> |-|--|--|--|--|--|
> | Before unlearning        | 45.8                  | 51.8                     | 49.6              | 55.1              | 56.6                       |
> | After unlearning (PPO)   | 47.6                  | 54.2                     | 52.1              | 57.4              | 59.2                       |
> | After unlearning (ReMax) | 48.5                  | 53.9                     | 51.5              | 57.6              | 58.4                       |
>
> The above results show that the reward models become better after unlearning.

---

> > ### Comment · Reviewer_Xdyj · 2025-08-06
> >
> > Thanks for the author’s rebuttal. The lack of comprehensive experiments on recent models of sufficient size is the reason preventing a higher score. I will retain my original score.

---

> > > ### Author Response · Authors · 2025-08-06
> > >
> > > We sincerely appreciate your time and effort in reviewing our paper and thank you for recognizing the contributions of this work. For your information, we would like to mention that in addition to the experiment on the newer Llama-3.1-8B model, we also include an experiment on a larger llama-2-13b model. We include the experiment results in our response to Reviewer 66Se, and we would like to provide those results here:
> > >
> > > We provide win rates (over the sft model) performance of llama-2-13b base model on imdb dataset. The win rates are evaluated by GPT-4.
> > >
> > > | PPO  | PPO+XRLHF | ReMax | ReMax+XRLHF |
> > > |------|-----------|-------|-------------|
> > > | 69.6 | 72.5      | 70.7  | 73.2        |
> > >
> > > The above results validate the effectiveness of our method on a larger 13B model.

---

### Official Review · Reviewer_112q · 2025-07-02

**Clarity:** 3
**Significance:** 3
**Originality:** 4
**Rating:** 5
**Confidence:** 3

**Summary:**

The paper introduces XRLHF, an approach that aims to further improve an LLMs adherence to preferences *after* initial preference fine-tuning using a dataset of "unsatisfactory responses".
The method starts by identifying preferences from the training data that encourage these unsatisfactory responses (which they show is done by training data that can be convexly combined to closely match the unsatisfactory sample) and then proposes an approach to unlearn these problematic preferences.
The unlearning is done in two steps:
First, the reward model that was used in the initial preference fine-tuning is trained with inverse gradients to unlearn the undesirable preferences.
Then, the language model is fine-tuned on this updated reward model, but only on the unsatisfactory responses under the constraint of keeping the KL-divergence on another set of satisfactory responses small.

The authors empirically evaluate this approach on two datasets (chat and summarization) and with two RLHF backbones. They show that with 500 validation samples (labeled to be satisfactory or unsatisfactory; around 25-30% of them are unsatisfactory) they can significantly improve LLM performance. Both labeling and evaluation is done by an external reward model (different from the one used in the initial RLHF training, likely with greater capability).

The key contributions are (1) an approach to identify training preference data that encourages a response; (2) a preference data unlearning approach; (3) a sample-efficient secondary preference fine-tuning approach (the combination of (1) and (2)).

**Questions:**

Please see the numbered weaknesses, focusing on the major ones (**WX**). Most of them are actionable, i.e., can be considered questions, although many of the minor ones (**MX**) do not require an individual response. Actual questions regarding the understanding of the paper are listed as weaknesses in the clarity section. The minor points have little impact on my score.

I think the contribution is technically solid (leading to my score on the acceptance side), but further clarification and more thorough evaluation would improve it. I consider W1, W7, and W9 to be of particular importance. Note that it is not necessary to conduct *all* experiments I suggest in the Quality section, but *some* comparison with baselines, particularly natural ones like direct unlearning, would greatly improve the paper.

**Ethical Concerns:**

["NO or VERY MINOR ethics concerns only"]

**Final Justification:**

The reviewers have addressed my concerns with extensive new experiments and clarifications. I believe their method would be a valuable contribution to NeurIPS. No major issues remain unresolved. Further details can be found in my initial review and the response to the authors.

**Limitations:**

- **W11** The discussion of the limitations is slightly limited. One major limitation, which I think is overlooked, is that as models get better, the fraction of unsatisfactory responses should get smaller. The rate of unsatisfactory responses in the paper seems high, likely due to the small models used. A lower fraction would require a larger validation set, making the method less applicable as models get better.

**Paper Formatting Concerns:**

- **N** I think $D$ in line 661 should be $D_u$.

**Quality:**

2

**Strengths And Weaknesses:**

In the following, I discuss strengths and weaknesses on the dimensions of quality, clarity, significance, and originality. Weaknesses are marked with a **W**, minor weaknesses with an **M**, notes that do not impact my score with **N**, and strengths with an **S**. Weaknesses are numbered for reference purposes.

## Quality

- **W1** My understanding is that your key contribution is a data-efficient method to further improve language models after initial fine-tuning. The empirical evidence supporting this data-efficiency is somewhat lacking, however, as there are no comparisons to any other approaches. Possible alternatives include:

	- Use direct unlearning methods to unlearn D_u in an SFT manner (I consider this the most important baseline, maybe the most important of all the weaknesses).

	- Add 500 (size of D) more samples to the initial RLHF dataset. This is not entirely equivalent to what you're doing, since RLHF is based on pairwise comparisons and you require binary labels. I believe it is quite comparable in effort, however. Alternatively perform an additional round of RLHF with only 500 samples (heavily KL-regularized / small learning rate / one epoch).

  To properly understand the effectiveness of your approach, I think it would be important to compare to baselines like these. I understand that this is computationally expensive and I believe it would be sufficient to do it on one algorithm/dataset combination. Still, some baseline on how large the improvement of "traditional methods" with 500 additional data points would be would be interesting and important to evaluate your approach.

- **W2** It would be interesting to see a comparison to a baseline that does a full retraining run without the "bad" preference data. As you say, this is more computationally expensive than your method. It would still be valuable to see how it compares in terms of performance, at least in one experimental condition (one backbone, one dataset).

- **W3** On line 3678 you claim that since win rates on D are smaller than those on D_u, this implies degradation on D \ D_u. This is not necessarily true, win rates would also be smaller if performance on D \ D_u remains unchanged. Can you report performance on D \ D_u separately to quantify the degradation?

- **W4** (This is lower priority, considering that is likely a significant effort).
You claim that a major differentiator of your approaches from other approaches aiming for alignment with noisy preference data is that it also works with misleading preferences that are valid for some prompts but not for others. I am not entirely convinced that this is practically relevant for many cases (as the prompt is part of the RM training data and a sufficiently a well-trained RM should be able to distinguish cases where the preference is relevant and cases where it is not). Either way, it would be possible to empirically compare your method to alternative approaches. PerpCorrect (your [14]) seems particularly relevant here, as it does not need any additional data. Your method could be combined with PerpCorrect, however it is imaginable that the validation dataset would not contain sufficient "bad" examples after applying PerpCorrect, rendering your method ineffective. I would therefore like to see how the performance of PerpCorrect compares to the performance of your method (standalone and/or combined with PerpCorrect).

## Clarity

- **S** The paper is generally well-written and clear. I especially enjoyed the clarity of the preliminaries section.

- **W5** It is not clear to me how the quadratic program in equation 2 is solved in practice.

- **W6** Are the weights of the convex combination entirely ignored in the unlearning phase? If so, is this reasonable? This should be discussed.

- **W7** As far as I understand it, the "Remark on general reward cases" is only valid if you consider the representation part to be frozen and only train the last linear layer during fine-tuning. Is that the case?

- **W8** You use reward models for evaluation and labeling. I think this is okay for research purposes, but not realistic in practice: If you had a sufficiently good reward model available, you could just train on it directly. This should be discussed in the paper.

- **W9** Why is it desirable that each datapoint of the composition is close to the "bad" training data point? Wouldn't it be better to have the smallest possible decomposition instead?

- **W10** Doesn't minimizing the LL of The example in Table 8 then encourage the Khaki response, which is not necessarily better? The core problem is the misunderstanding of the prompt.

- **M1** In the worst case in Algorithm 1, it may happen that the entire dataset has to be considered. Does it have any benefit for the worst-case complexity to iterate over the dataset starting from the closest element?

- **M2** Why are there no examples of no XRLHF in D.4? That would be more interesting than PPO/ReMax comparison.

- **M3** You claim that the unlearning reduces the influence of S (line 244). Would it not be more correct to say that it *inverts* the influence?

- **M5** The claim on line 142 (policy generates response -> response must have had high reward) is not necessarily always true, as RLHF training uses only limited optimization and a KL constraint. It's still "true enough" for your purposes, but the claim should be slightly weakened.

- **M6** I was initially confused by line 186 (I think there are more instances of this) claiming to find the convex combination that is closest to the projected target. I assumed this meant the actual combination is close, which confused me as an exact match is ensured by the projection. What is actually meant is that each individual member of the decomposition is close. This becomes clear in the formula, but should be clarified in the text.

- **M7** On line 227 it is implied that there exists a single unique subset that "leads the policy model" to make an unsatisfactory response. Similarly later a causal relationship is implied. Both of these claims should be weakened.

- **N** The introduction discusses the method in quite some level of detail. That is not a problem, but it left me confused on two points: (1) How is it ensured that the decomposition exists? (Later clarified, projection); (2) How is such a decomposition even possible, as preference data differs in format from prompt-response pairs (also later explained). If this detailed description of the method remains in the introduction, those two points should be briefly clarified. In general, I think discussing early that you mostly work on the difference of response features would clarify things a bit.

- **N** You claim that "This idea is in the same spirit as DPO" (line 232). The same sentence would apply to RLHF, however. So I do not understand which value it adds.

- **N** The name XRLHF is used for the first time in section 6 and never formally introduced.

## Significance

I think the approach to generate preference-data based explanations and to unlearn preference data is significant. The combination of these two, potentially leading to a data-efficient secondary preference fine-tuning approach, is not evaluated and compared with (simpler) alternatives sufficiently thoroughly to determine its significance, however.

## Originality

The work provides original methods to the best of my knowledge. My familiarity with the relevant literature is somewhat limited, however. The authors describe related work and clearly distinguish themselves from it.

---

> ### Author Rebuttal · Authors · 2025-07-31
>
> Thanks for your detailed reviews. We address your comments in a point-by-point manner. Due to space limitation, we cannot include our answers to the **N** comments.
>
> **Answer to W1**: We add three baselines using the opt-1.3B model and full-hh-rlhf dataset: (1) DU: we directly unlearn $D_u$ from the RLHF policy $\pi_0$ by minimizing the log-likelihood of $D_u$; (2) RLHF500: we use the last 500 preference data in the test split of full-hh-rlhf, add this 500 data to the original train split, and run RLHF on the new train split. (3) RL500: we use the original reward model and original RLHF policy $\pi_0$ to run a second-round RL on the 500 data. Note that we exclude the 500 data when we evaluate performance on the test set. Therefore, the test set we use here is 500 examples smaller than the one used in the main paper. We report GPT-4 win rates of the original RLHF, our method, and the three new baselines over the SFT model on both the validation set $D$ and the new test set.
>
> |                | PPO  | PPO+XRLHF | PPO+DU | PPO+RLHF500 | PPO+RL500 | ReMax | ReMax+XRLHF | ReMax+DU | ReMax+RLHF500 | ReMax+RL500 |
> |-|-|-|-|-|-|-|-|-|-|-|
> | validation set | 68.4 | 78.1      | 74.1   | 69.3        | 71.1      | 69.2   | 77.1         | 73.8     | 70.1           | 70.6         |
> | test set       | 68.1 | 77.2      | 73.4   | 68.8        | 70.2      | 67.3   | 75.8         | 72.9     | 68.8           | 68.2         |
>
> The above results show that our method (XRLHF) significantly improves RLHF on both validation and test sets, while RLHF500 and RL500 only slightly improve RLHF. The method DU also improves RLHF, however, the improvement is smaller than XRLHF. The reason is that unlearning unsatisfactory responses can degrade satisfactory responses (explained in lines 237-239). In contrast, XRLHF not only unlearns the unsatisfactory responses but also retains the satisfactory responses (explained in lines 257-260).
>
> **Answer to W2**: We compare our method (XRLHF) to full retraining (FR) using the opt-1.3B model and full-hh-rlhf dataset. Specifically, we exclude the explanation (bad preference data) from the original reward training data, retrain a reward model on the new reward training data, and use the retrained reward to run RL. We consider two variants of FR: (1) naive FR: we follow the standard RLHF pipeline and use the retrained reward to run RL on the SFT model. (2) retain FR: we follow lines 257-263 where we use the retrained reward to tune the original XRLHF policy only on the unsatisfactory responses but restricts KL divergence from the original XRLHF policy on the satisfactory responses. We report GPT-4 win rates on both validation and test sets. The test set is the one used in the answer to W1.
>
> |                | PPO+XRLHF | PPO+naive FR | PPO+retain FR | ReMax+XRLHF | ReMax+naive FR | ReMax+retain FR |
> |-|-|-|-|-|-|-|
> | validation set | 78.1      | 73.2         | 78.5          | 77.1        | 74.0           | 77.3            |
> | test set       | 77.2      | 74.1         | 77.9          | 75.8        | 72.5           | 75.2            |
>
> The above results show that our method outperforms naive FR because naive FR can degrade satisfactory responses (explained in lines 237-239). The retain FR performs slightly better than our method because it completely removes the effect of the explanation and retains the satisfactory responses (lines 257-260). However, our reward unlearning achieves similar performance while requiring around 8\% of the time for reward retraining.
>
> **Answer to W3**: To quantify the degradation on $D\setminus D_u$, we provide the percentages of prompts of $D\setminus D_u$ where XRLHF response is worse than RLHF response (evaluated by GPT-4).
>
> |                         | PPO+XRLHF worse than PPO | ReMax+XRLHF worse than ReMax |
> |-|-|-|
> | opt+full-hh-rlhf        | 32.7%                    | 41.4%                         |
> | pythia+full-hh-rlhf     | 29.2%                    | 31.0%                         |
> | pythia+TL;DR            | 33.2%                    | 30.8%                         |
> | llama+TL;DR             | 27.9%                    | 25.6%                         |
>
> **Answer to W4**: We compare our method to PerpCorrect based on opt-1.3B model and full-hh-rlhf dataset. Since PerpCorrect does not have ReMax implementation, we only compare to it using PPO as the base RLHF algorithm. We provide the GPT-4 win rates of our method over PerpCorrect on both the validation and test sets: 60.5\% (validation set), 56.2\% (test set). Since the win rates are above 50\%, it shows that our method outperforms PerpCorrect in general.
>
> **Answer to W5**: We use a python package "cvxpy" to solve this quadratic program. First, we need to specify the variable, objection function, and constraint in equation 2. Second, we can formulate the quadratic program using "problem=cvxpy.Problem(objective,constraint)" and solve this problem using "problem.solve()".
>
> **Answer to W6**: The weights of the convex combination are not ignored in the unlearning phase. Since a convex combination can include many training data with negligible weights, in practice, we only pick the top 30\% data with highest weights as explanation and only unlearn these 30\% data.
>
> **Answer to W7**: Your understanding is correct. During reward unlearning, we only update the parameters of the final linear layer, while keeping the representation part frozen. This design has two benefits: (1) interpretability: since the representation part is fixed, it is easy for humans to interpret how a test-time example can be decomposed by training examples and how much each training example contributes to the high reward of the test-time example; (2) efficiency: only updating the final linear layer during unlearning is much more efficient than updating the parameters of both the transformer and final linear layer.
>
> **Answer to W8**: We will add a discussion that if a sufficiently good reward model is available, a practical approach is to directly train a policy on it. In our case, we do not assume the access to such a good reward model during training. Instead, this kind of reward model is only used as a proxy for human judgments during evaluation.
>
> **Answer to W9**: We agree that finding the smallest possible decomposition can also be a valid solution. However, this approach can be computationally intractable in practice, because it requires searching over all possible subsets of the training data (which is a space that grows exponentially with the training data size) to find the smallest decomposition. In contrast, our method finds the closest decomposition, which enables an efficient search strategy: we start from the nearest neighbor and iteratively add new data until a decomposition is found. This strategy is much more efficient because it avoids searching over all the possible decompositions (explained in lines 201-207), and we prove that this method's computation complexity is only polynomial (Theorem 1). Empirically, we find that it is typical to add no more than 100 nearest data points until a valid decomposition is found, while the total number of reward training data is more than 35,000.
>
> **Answer to W10**: The goal of reward unlearning is to eliminate the effect of the example in Table 8 on the reward model, not to encourage the rejected response of this example. Since the original reward model is learned by maximizing the log-likelihood of this example, we use negative gradient to unlearn this effect. We agree that fully minimizing the log-likelihood of this example can encourage the rejected response. However, we do not run the negative gradient until convergence, instead, we only run several steps of negative gradient to unlearn this example.
>
> **Answer to M1**: In the worst case where the entire dataset needs to be considered, starting from the closest data point does not have additional benefit. However, it is beneficial in practice because empirically, we find that a valid decomposition is typically found by considering no more than 100 nearest data points, while the total number of the data points is more than 35,000.
>
> **Answer to M2**: We will add responses generated by PPO/ReMax. In D.4, we only include examples of XRLHF because we want to demonstrate the responses generated by our algorithm, and the comparison of XRLHF and RLHF is already evaluated using win rates.
>
> **Answer to M3**: The goal of unlearning is not to invert the influence but to reduce the influence. We agree that fully minimizing the likelihood can invert influence, however, we only deploy several steps of negative gradient to reduce the influence.
>
> **Answer to M5**: We will add a discussion to revise this claim and clarify that this claim considers a simplified case, while in practice, the policy may not maximize the reward because of the insufficient RL optimization and the KL constraint.
>
> **Answer to M6**: We will add a clarification to the text that the closest convex combination means the combination where the sum of the distance of each member to the projected target is minimum.
>
> **Answer to M7**: We will clarify that multiple decompositions may exist to influence the LM's responses and our method just finds one of them. We will also discuss that our explanation provides a plausible attribution of influence on the LM's responses, but there could be other sources that also influence the LM's responses.
>
> **Answer to W11**: We will add the limitation that for larger models with high capability, in order to get enough unsatisfactory examples to explain, our algorithm may need a larger validation set.

---

> > ### Comment · Reviewer_112q · 2025-08-04
> >
> > Thank you for your detailled response. The new experiments (especially for W1 and W2; I appreciate that you also included the non-naive variant for W2) and clarifications have addressed my concerns.
> >
> > I think the paper would be improved if you add these clarifications to the final revision, especially the clarifications for W6 (should still be discussed that within the chosen examples, weight no longer matters), W7 (should be made explicit that layers are frozen) and W9 (should be discussed).
> >
> > I will raise my score to recommend acceptance (5).

---

> > > ### Author Response · Authors · 2025-08-04
> > >
> > > We sincerely appreciate your time and effort in reviewing our paper and thank you for recognizing the contributions of this work. We will include the additional experiments and clarifications on W6, W7 and W9 in the new version of the paper.

---

### Note · Authors · 2025-08-11

Dear AC and Reviewers,

We would like to take this opportunity to summarize our discussions with the reviewers and appreciate the reviewers’ effort and engagement in reviewing our paper. We really appreciate that all the reviewers recognize the contributions of this paper. Specifically, three reviewers (Reviewer 112q, Reviewer 8SuZ, Reviewer 66Se) are satisfied with our discussion and decide to increase the rating. One reviewer (Reviewer Xdyj) decides to maintain the original positive rating, and in response to Reviewer Xdyj, we provide an additional experiment result on a larger 13B model.

We will follow the reviewers' suggestions and include the additional experiment results and discussions in the paper.

Best,
Authors

---

### Decision · Program_Chairs · 2025-09-17

**Decision:**

Accept (poster)

**Comment:**

This a post-RLHF “explain-and-correct” method that identifies preference training data responsible for unsatisfactory model outputs and then performs targeted unlearning to improve alignment. The idea is novel: the method formulates explanation as finding convex decompositions over reward data, and correction as inverse-gradient unlearning of these problematic preferences, followed by constrained fine-tuning. The authors argue that this approach is more data-efficient than retraining and can be applied in a lightweight manner to improve existing RLHF policies.

Strengths of the paper by the reviewers;
- Novelty: The paper introduces a framework to link unsatisfactory responses back to preference training data, then uses this link to unlearn harmful signals. This is an original contribution and resonates with broader themes in explainability and post-training alignment.
- Practical value: As a post-training step, XRLHF is attractive for deployed or expensive-to-train models, making it potentially impactful in applied settings.
- New experiments in rebuttal: The authors provided substantial additional experiments, including ablations (e.g., proportion of data unlearned, KL trade-offs), larger models (LLaMA-3.1-8B, LLaMA-2-13B), generalization benchmarks (AlpacaEval, SafeRLHF, Webis TL;DR-17, IMDb), and reward model performance before and after unlearning. These additions convincingly strengthen the paper.

Some weaknesses were raised by the reviewers:
- Model scale: Some reviewers noted reliance on smaller models in the main paper; while newer results with LLaMA-3.1-8B and LLaMA-2-13B mitigate this, broader evaluation on strong modern models would further increase confidence.
- Explainability framing: Multiple reviewers flagged that “explainable RLHF” may overstate the level of interpretability. The authors clarified that explanation is used as a tool for improvement, provided causal reasoning pathways, fidelity/human plausibility evaluations, and plan to revise the title/abstract.

The reviewers agree toward acceptance after rebuttal. Concerns about baselines, scalability, and clarity of the “explainability” claim have been meaningfully addressed with new experiments and clarifications. The technical contribution is sound, original, and practically relevant, and the paper is also well-written.